# Introduction of a More Glutaredoxin-like Active Site to PDI Results in Competition between Protein Substrate and Glutathione Binding

**DOI:** 10.3390/antiox11101920

**Published:** 2022-09-28

**Authors:** Mirva J. Saaranen, Heli I. Alanen, Kirsi E. H. Salo, Emmanuel Nji, Pekka Kärkkäinen, Constanze Schmotz, Lloyd W. Ruddock

**Affiliations:** 1Protein and Structural Biology, Faculty of Biochemistry and Molecular Medicine, University of Oulu, 90220 Oulu, Finland; 2Research Unit of Biomedicine, Faculty of Medicine, University of Oulu, 90220 Oulu, Finland; 3Centre for Research in Therapeutic Sciences (CREATES), Strathmore University, Nairobi 59857-00200, Kenya; 4BioStruct-Africa, Vårby, 14343 Stockholm, Sweden; 5Research Program for Clinical and Molecular Metabolism, University of Helsinki, 00290 Helsinki, Finland

**Keywords:** protein disulfide isomerase, glutathione, glutaredoxin, thioredoxin fold, disulfide bond, redox

## Abstract

Proteins in the thioredoxin superfamily share a similar fold, contain a -CXXC- active site, and catalyze oxidoreductase reactions by dithiol-disulfide exchange mechanisms. Protein disulfide isomerase (PDI) has two -CGHC- active sites. For in vitro studies, oxidation/reduction of PDI during the catalytic cycle is accomplished with glutathione. Glutathione may act as electron donor/acceptor for PDI also in vivo, but at least for oxidation reactions, GSSG probably is not the major electron acceptor and PDI may not have evolved to react with glutathione with high affinity, but merely having adequate affinity for both glutathione and folding proteins/peptides. Glutaredoxins, on the other hand, have a high affinity for glutathione. They commonly have -CXFC- or -CXYC- active site, where the tyrosine residue forms part of the GSH binding groove. Mutating the active site of PDI to a more glutaredoxin-like motif increased its reactivity with glutathione. All such variants showed an increased rate in GSH-dependent reduction or GSSG-dependent oxidation of the active site, as well as a decreased rate of the native disulfide bond formation, with the magnitude of the effect increasing with glutathione concentration. This suggests that these variants lead to competition in binding between glutathione and folding protein substrates.

## 1. Introduction

Proteins belonging to the thioredoxin superfamily share a similar fold and catalyze oxidoreductase reactions by dithiol-disulfide exchange mechanisms. They contain a -Cys-X-X-Cys- active site motif. The reactions catalyzed by individual members are determined by the redox equilibrium of the disulfide-dithiol state of the active site cysteines and their substrate specificity [1]. Thioredoxin superfamily members include thioredoxins, glutaredoxins, and protein disulfide isomerases (PDI). Canonical human PDI contains two active sites with -CGHC- motifs. It catalyzes disulfide bond formation in the endoplasmic reticulum (ER) via oxidation and isomerization reactions [2]. For in vitro studies, oxidation/reduction of PDI during the catalytic cycle is usually accomplished with glutathione. Often, a redox buffer comprising both oxidized glutathione (GSSG) and reduced glutathione (GSH) is used. The pathways for disulfide bond formation in vivo, however, are complex and not completely understood. There exist multiple pathways for disulfide formation [3,4], with the primary physiological route being thought to be via the action of the sulfhydryl oxidase Ero1, which oxidizes the active site of PDI. PDI, in turn, transfers that disulfide to folding proteins, becoming reduced in the process.

Since the glutathione redox potential in the ER is more oxidizing than in the cytoplasm [5] and is close to the optimal redox potential used in vitro to refold proteins, it is likely that glutathione acts as an electron donor/acceptor for PDI in vivo. However, at least for oxidation reactions, GSSG probably is not the major electron acceptor, and PDI may not have evolved to react with glutathione with high affinity since this could lead to futile cycling. Rather what would be expected would be that PDI has an appropriate affinity for both glutathione and folding proteins/peptides to enable efficient utilization of the glutathione redox potential catalyzing protein folding.

Glutaredoxins, on the other hand, have a high affinity for glutathione and primarily catalyze deglutathionylation reactions, i.e., reduction of protein-GSH mixed disulfides [6]. Both PDI and glutaredoxins are members of the thioredoxin superfamily, both share the same fold (Appendix A, [7,8]) but their active sites differ. Glutaredoxins commonly have a tyrosine or phenylalanine prior to the C-terminal active site cysteine in their -Cys-X-X-Cys- active site, i.e., CXFC or CXYC. This tyrosine residue has been found to form part of the GSH binding groove [9,10].

In this study, the PDI active site motif was mutated to contain either phenylalanine or tyrosine in the place of histidine to make it more glutaredoxin-like. These mutations did not cause any gross structural defects, as determined by far-UV CD. Mutating the PDI active site to more glutaredoxin-like motif increased its reactivity with glutathione. All such variants showed an increased rate in GSH-dependent reduction or GSSG-dependent oxidation of the PDI active site, as determined by stopped-flow experiments, as well as the ability to catalyze deglutathionylation reactions. The effect of variants on the ability to catalyze both oxidative foldings (oxidation and isomerization reactions) was studied using bovine pancreatic trypsin inhibitor (BPTI) refolding assays. Variants that increased the rate of reaction with glutathione decreased the rate of native disulfide bond formation, with the magnitude of the effect increasing with glutathione concentration. The results suggest that generating a more glutaredoxin-like active site in PDI leads to a competition between binding to glutathione and to folding proteins. These results are consistent with a functional role for glutathione in physiological native disulfide bond formation in the ER with glutathione acting as a redox buffer and that PDI has evolved to have an adequate and appropriate affinity for both glutathione and folding proteins.

## 2. Materials and Methods

### 2.1. Protein Production and Purification

All constructs used are listed in Table 1. Mutant forms of mature human PDI, as well as the catalytic **a** domain of human PDI, were generated by site-directed mutagenesis according to the instructions of the QuikChange Site-Directed Mutagenesis protocol (Agilent, Santa Clara, CA, USA). All genes of interest cloned into vectors were checked for correctness by DNA sequencing. Proteins were expressed in the *E. coli* strain BL21(DE3) pLysS grown in LB medium at 37 °C and induced at an A_600_ of 0.3 for three hours with 1 mM IPTG. Expressed PDI variants were purified by immobilized metal affinity chromatography and ion exchange chromatography as described for the **a** domain of PDI [11]. Pure fractions, as determined by Coomassie Brilliant Blue-stained SDS-PAGE, were combined and buffer-exchanged into 20 mM sodium phosphate buffer, pH 7.3, and stored frozen at −20 °C. BPTI was purified as described previously [12]. The concentration of each protein was determined spectrophotometrically using a calculated molar absorption coefficient. All purified proteins were analyzed for quality and authenticity, as determined by expected molecular weight, by liquid chromatography mass spectrometry (LC-MS) as described in Gaciarz and Ruddock (2017) [13].

### 2.2. Circular Dichroism

Far-UV circular dichroism (CD) spectra were recorded with a Chirascan Plus CD Spectrometer (Applied Photophysics, Surrey, UK) as described previously [16]. The HT voltage did not exceed 750 V. At least three replicates for each construct were recorded.

### 2.3. BPTI Refolding 

The BPTI refolding was carried out as described previously [16] with the exceptions that the buffer used was McIlvaine buffer pH 6.5 and glutathione buffer was either with 0.5 mM GSSG and 2 mM GSH or 1 mM GSSG and 8 mM GSH.

### 2.4. Kinetics of Oxidation and Reduction 

Before the assays described below, the proteins were either reduced with a 10-fold molar excess of DTT or oxidized with a 10-fold molar excess of GSSG for 30 min at room temperature, followed by removal of the excess of DTT/GSSG and buffer exchange into desired reaction buffer using centrifugal filter device (Amicon Ultra with MWCO 10 K).

The rates of oxidation and reduction of the active site of different PDI **a** domain constructs by glutathione were determined using a KinTek SF-2004 stopped-flow fluorometer (KinTek, Snow Shoe, PA, USA) as described previously [11] except a final concentration of 10 μM of enzyme and 0–15 mM GSH or 0–40 mM GSSG were used.

The rate of oxidation by DHA was determined as described previously [17] (Saaranen et al., 2009) with 20 µM of the reduced enzyme.

### 2.5. Ellman’s Assay 

The reactivity of active site cysteines of the different PDI **a** domain constructs towards Ellman’s reagent (DTNB) was analyzed by measuring the change in absorbance at 412 nm using KinTek SF-2004 stopped-flow instrument as described previously [18] in 0.1 M Na_2_HPO_4_, 0.1 M citric acid, 0.1 M boric acid, 1 mM EDTA, pH 7.0 with 184 µM DTNB and 10 µM reduced enzyme.

### 2.6. Deglutathionylation Assay 

The deglutathionylation activity of the enzymes was analyzed based on a method described previously [19]. Briefly, fluorescence measurements were carried out with FluoroMax-4 (HORIBA Instruments, Irvine, CA, USA) in McIlvaine buffer (0.2 M disodium hydrogen phosphate, 0.1 M citric acid) at pH 7.0, including 1 mM GSH, 50 µM NADPH, 0.06 U glutathione reductase, 1 mM EDTA, 1 µg/mL BSA, 5 µM substrate peptide, and 200 nM enzyme of interest. All measurements were done at 25 °C, excitation 280 nm with slit width 1 nm and emission 356 nm with slid width 5 nm.

## 3. Results

### 3.1. Identification of Variants with Increased Reactivity towards Glutathione

As part of the renewal of the Biochemistry BSc curriculum at the University of Oulu the practical components of several second-year courses were linked together to provide the students a more comprehensive overview of workflows in biochemical research. In *Molecular Biology I,* students generated individual point mutations in the active site of an enzyme. In *Protein Chemistry I,* they purified their protein variant. In *Biochemical methodologies II,* they characterized the activity of the variant and undertook biophysical characterization, and in *English for Biochemists II,* the students compiled the data from all of the variants and wrote a report in the style of a scientific publication. Neither students nor staff would know the expected results as the mutations made were random. The enzyme chosen was the catalytic **a** domain of human PDI, as it was expected that a large number of different variants could be obtained that decreased one or more activities. Furthermore, it was anticipated that these variants could be clustered by the students during their analysis e.g., mutations affecting accessibility or reactivity or pK_a_ of the N-terminal active site cysteine, in mutations affecting pK_a_ of the C-terminal active site cysteine or mutations which destabilized the protein, etc. While large numbers of variants with such expected activity modulating effects were obtained, unexpectedly, several variants were obtained, which increased the activity of this domain in oxidative folding of a peptide substrate [20] and in reactivity towards GSSG and GSH. These variants were G54N/H55F, G54P/H55F, G54Q/H55F, and G54S/H55F. All had the common feature of having an aromatic residue replacing the histidine in the WCGHC active site motif.

To examine this effect, more detailed characterization of these variants along with the single point mutations H55F and H55Y were made in the isolated **a** domain as well as the double active site mutations H55F/H399F and H55Y/H399Y in full-length mature human PDI. All variants were analyzed by far UV circular dichroism, and no significant differences were observed in the spectra compared to that of the wild-type protein (Figure 1 and Appendix A). This implies that these mutations do not induce gross structural changes in the proteins.

The rates of reaction of the **a** domain constructs with GSSG and GSH were examined by redox-dependent changes in fluorescence of the tryptophan in the WCXXC active site. This was facilitated by all constructs having the W128F mutation to remove the other tryptophan in the protein [14]. Oxidation of the reduced active site by GSSG is a two-step process (Figure 2A), nucleophilic attack by Cys53 to form a mixed disulfide, and subsequent nucleophilic attack by Cys56. The second step is modulated by movement of Arg120 [14]. The fluorescence change associated with the second step is small (maximally 10% of the total change), and the magnitude is dependent on buffer composition and variant of the active site. It cannot be observed in most cases tested here. The first step is dependent on [GSSG], and with GSSG in large excess, the fluorescence change fits a pseudo-first-order event (Figure 2B). A linear plot of the determined first-order rate constant vs. [GSSG] allows the determination of the second-order rate constants (Figure 2C). All six variants tested where His55 was mutated to tyrosine or phenylalanine showed a significant increase in the rate of reaction with GSSG compared to wild-type protein, with the maximal rate being more than two-fold faster (Figure 2D). Significant variations were seen between the variants, with the G54S/H55F variant being the fastest and G54P/H55F being the slowest.

Reduction of the oxidized active site by GSH is also a two-step process (Figure 2A), with nucleophilic attack by GSH being involved in both steps. The kinetics of reduction are complex [11] due to the reverse reaction. At high concentrations of GSH the [GSH] dependence of the rate is linear, but at low concentrations of GSH the rate is proportional to [GSH]^2^, in effect, the reverse reaction is so fast that the overall reaction is third order. A plot of the determined pseudo-first-order rate constant for reduction by GSH against [GSH]^2^ was linear for all variants tested and allows the determination of the third order rate constants (Figure 2E). All six variants tested show a significant increase in the rate of reaction with GSH compared to wild-type protein, with the maximal rate being circa eight-fold faster (Figure 2F). Significant variations were seen between the variants, with the G54P/H55F variant being the fastest and the G54S/H55F variant being the slowest.

### 3.2. Specificity of the Variants towards Glutathione

A variety of inter-dependent factors affect the rate of oxidation and reduction of the active site of PDI, including redox potential, pK_a_ of Cys53 and Cys56, accessibility, and the affinity for the oxidizing and reducing agents used. For any individual variant, combinations of these factors may be involved. However, both the initial step of oxidation by GSSG and the rate of reduction by GSH increased in all the variants and the only common factor that would increase both of these rates would be either accessibility of the active site or the affinity for glutathione.

To discriminate between these two options, the reactivity of the active site of the variants towards other chemical species was examined. The reaction of the reduced **a** domain with excess dithionitrobenzoate (DTNB; Ellman’s reagent) was monitored by stopped flow using changes in absorbance at 412 nm [18]. As expected, all reactions fitted pseudo-first-order reactions (Figure 3A). While some of the variants showed a significant increase in relative activity, others showed a significant decrease (Figure 3B). The maximal changes were less than observed for the reaction with GSH/GSSG, and no correlation was observed between the relative rates of reaction with DTNB and GSSG or GSH (Appendix A). From here on, we focused on the active site variants where only the histidine was changed either to phenylalanine or tyrosine. While the H55Y and H55F variants showed an increased rate of oxidation by GSSG (Figure 2D), both variants showed a decreased rate of oxidation by dehydroascorbate (DHA; Figure 3C). Combined, these results imply that most likely, the variants increased the reaction rates with glutathione by increasing the affinity for glutathione.

Increasing the affinity of the active site of PDI for glutathione would be predicted to increase the activity of PDI catalysis of deglutathionylation reactions. Using changes in fluorescence of a model glutathionylated peptide ([19]; Figure 3D), the relative rates of reaction were determined, and as expected, PDI with both active sites mutated to either H55F/H399F or H55Y/H399Y showed a significant increase in deglutathionylation activity compared with the wild-type enzyme (Figure 3E).

### 3.3. Glutathione Acts As a Competitor for Binding Protein Substrates

To examine the effect of the mutations on other PDI activities, we examined the refolding kinetics of the model protein bovine pancreatic trypsin inhibitor (BPTI) by mass spectrometry. Under standard refolding conditions, with 2 mM GSH and 0.5 mM GSSG, both the H55Y/H399Y and H55F/H399F variants behaved similarly, with both resulting in a slightly slower rate of attainment of the native three disulfide (3S) state of BPTI compared with the wild-type enzyme (Figure 4A). The kinetics of BPTI folding are complex, with the formation of multiple folding intermediates [21,22]. However, it is possible to determine rate constants for the disappearance of the fully reduced state to form the one disulfide state (1S)—primarily the Cys30-51 disulfide is formed first [21]—and the disappearance of the 1S state to form the two disulfide (2S) state. Under standard conditions, the rate of formation of the 1S state is not significantly different between the wild-type and variants (Figure 4B). In contrast, the rate of formation of the 2S state is significantly slower for both variants (Figure 4C).

The reduced rate of catalysis of late-stage folding events of BPTI by the PDI variants is not due to reduced rates of redox-exchange with glutathione, as both variants showed increased rates of oxidation by GSSG and reduction by GSH (Figure 2). It is plausible that the increased affinity for glutathione reduces the interaction between PDI variants and BPTI folding intermediates, i.e., that glutathione acts as a competitor in the system. To test this hypothesis, BPTI refolding was repeated with higher concentrations of glutathione while keeping the redox potential of the GSSG/GSH buffer the same. Increasing glutathione concentrations should increase the rate of catalyzed oxidative folding unless glutathione is acting as a competitive inhibitor for substrate binding. At these higher glutathione concentrations, the rate of both non-catalyzed and wild-type PDI catalyzed reactions increased (Figure 4). In contrast to the results obtained under standard concentrations of glutathione (Figure 4B), there was a significant reduction in the rate of catalysis of the formation of the 1S species for both variants compared to wild-type (Figure 4E). In addition, while the rate of catalysis of formation of the 2S species increased significantly for wild-type PDI with increasing concentrations of glutathione (Figure 4C vs. Figure 4F; *p* = 0.003), neither variant showed a significant change in rate with increasing glutathione concentration.

## 4. Discussion

To function PDI must interact with a wide range of molecules, both low molecular weight and macromolecules. PDI catalyzes oxidation, reduction, and isomerization of disulfide bonds in folding substrates in the ER. Catalysis of reduction and isomerization requires the active site of PDI to be in the reduced state (dithiol), while in contrast, catalysis of oxidation requires the active site to be in the oxidized state (disulfide). Cycling between these two states is an essential part of the physiological function of PDI.

Oxidation of the active site of PDI can be done via a wide range of pathways ([2,4], including the action of Ero1 homologs [23,24], GSSG [11], Vitamin K oxidoreductase [25], recycling DHA back to ascorbate [17] and hydrogen peroxide either directly [26] or via the action of the PDI-peroxidases GPx7 and GPx8 [27] or the peroxiredoxin Prx4 [28].

In contrast, only three pathways are known to allow the reduction of the active site in vivo. One recently characterized pathway is based on transferring reducing equivalents derived from cytosolic thioredoxin system across the ER membrane via a still unknown transmembrane protein [29,30]. The other two are via disulfide transfer as part of oxidative catalysis of folding by PDI, i.e., transfer of the active site disulfide to a folding protein and via reduction of the active site of PDI by GSH. Hence, PDI must have an affinity for and reactivity towards GSH, which allows it to be reduced on a physiologically relevant timescale. Similarly, oxidation of PDI by GSSG is fast in vitro, and it would make sense for GSSG to be a physiologically relevant (if non-essential; [31]) oxidant for PDI in vivo. However, the intracellular concentrations of GSH, including in the ER, are high, though absolute quantification of GSH concentration and GSH:GSSG ratio in different organelles is difficult [5,32,33,34,35]. If PDI has a very high affinity for a highly abundant peptide like glutathione, then it will act as a competitor for binding folding protein substrates, preventing its physiological function. Similarly, too low an affinity for glutathione would inhibit its physiological function.

Here we report that variants in the active site, in particular mutations in the histidine in the CGHC motif to tyrosine or phenylalanine can increase the reactivity of the catalytic **a** domain of PDI towards GSSG up to 2.3-fold (Figure 2D) and towards GSH up to 8.4-fold (Figure 2B). These variants do not have a concomitant effect on oxidation by DHA (Figure 3B) or modulation of the reactivity towards DTNB (Figure 2A), and this, combined with the increased reactivity towards both GSH and GSSG, suggests that the effects are not due to modulation of the pK_a_ of either active site cysteine or due to altering their accessibility. Instead, the results suggest that these variants increase the affinity of PDI towards glutathione in both the oxidized and reduced states.

The effects of increased affinity towards glutathione would be predicted to be two-fold. Firstly, these variant enzymes would be expected to show increased affinity to glutathionylated species, increasing the rate of deglutathionylation reactions. Both the H55F and H55Y variants do show significantly increased rates of deglutathionylation catalysis (Figure 3E). Secondly, these variants might be expected to increase competition between the enzyme-binding glutathione and binding folding protein substrates. This also was observed (Figure 4). At low concentrations of glutathione typically used in in vitro experiments (2 mM GSH/0.5 mM GSSG) there was no significant difference in the rate of initial oxidation of reduced unfolded BPTI catalyzed by the wild-type enzyme in comparison with the H55F/H399F or H55Y/H399Y variants. In contrast, when higher absolute glutathione concentrations were used, the rate of initial oxidation of BPTI into 1S species was reduced with these variants compared to wild-type enzyme. Similarly, while the overall rate for formation of 2S species was increased for the wild-type enzyme, no effect was seen in the rate for the variants, implicating that the higher affinity for glutathione causes competition with folding substrates for the H55F/H399F and H55Y/H399Y variants.

## 5. Conclusions

It is reasonable to assume that PDI has evolved to have an adequate and appropriate affinity for both glutathione and folding protein substrates. The ER contains high levels of glutathione and creates an optimal redox potential for protein folding, at least in vitro. PDI needs to bind efficiently to the folding protein substrate, thus too high affinity for glutathione might be detrimental to its function, while too low would prevent it from utilizing the glutathione redox buffer. Another member of the thioredoxin superfamily, glutaredoxins, on the other hand, have a high affinity towards glutathione and glutathionylated substrates. In this study, we demonstrated that the introduction of a more glutaredoxin-like active site to PDI interferes with the optimal affinity of PDI towards different substrates by increasing the affinity towards glutathione, thus resulting in competition between binding glutathione and the folding protein substrate.

## Figures and Tables

**Figure 1 antioxidants-11-01920-f001:**
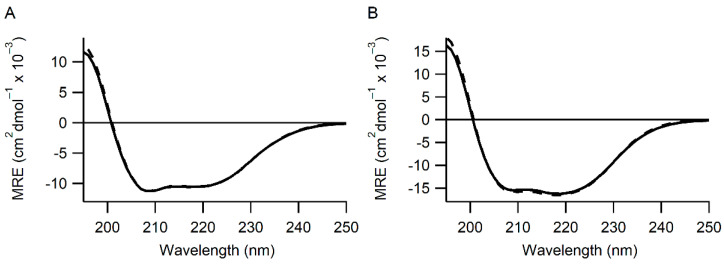
PDI variants have the same secondary structure as wild-type. Panel (**A**) Far UV CD spectra for wild-type (solid line) and H55Y H399Y (dashed line) mature PDI. Panel (**B**) Far UV CD spectra for wild-type (solid line) and H55Y (dashed line) **a** domain of PDI. Average traces of at least 3 separate CD spectroscopic scans are shown. MRE = mean residue ellipticity. All other variants tested showed similar CD spectra (Appendix A).

**Figure 2 antioxidants-11-01920-f002:**
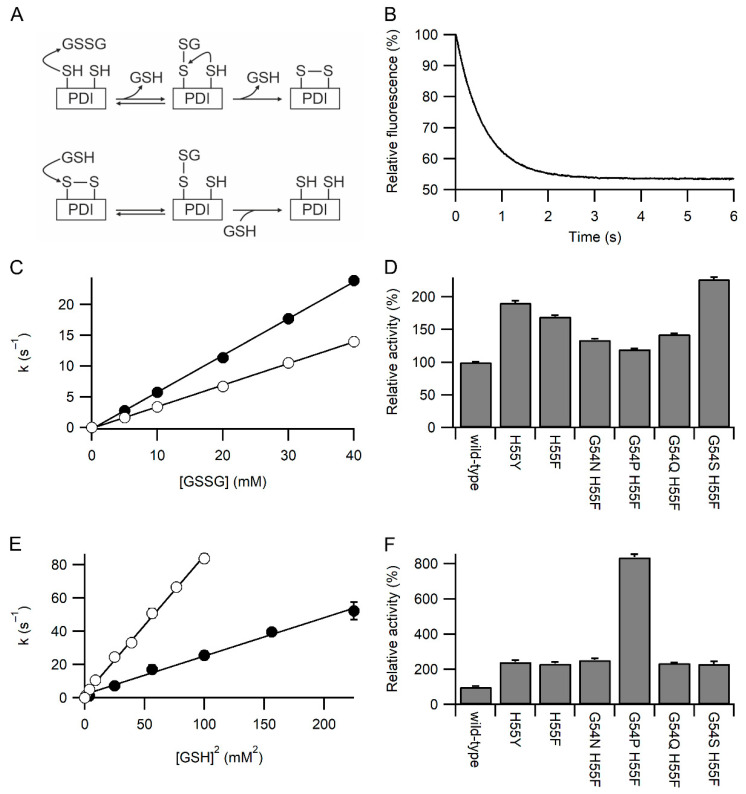
Kinetics of oxidation and reduction of the active site of PDI. Panel (**A**) Upper schematic of the reaction of reduced PDI active site with GSSG. The initial step gives pseudo-first-order reaction kinetics when GSSG is in excess and is responsible for >90% of the fluorescence change. Lower schematic of the reaction of oxidized PDI active site with GSH. Both steps give pseudo-first-order reaction kinetics when GSH is in excess. The rapid reverse reaction means that at low [GSH] the overall rate of reaction is proportional to [GSH]^2^, while at higher [GSH], it is proportional to [GSH] [11]. Panel (**B**) Representative plot for oxidation of wild-type PDI **a** domain with 5 mM GSSG showing the decrease in relative fluorescence over time. Panel (**C**) Linear dependence of the pseudo-first-order rate constant for oxidation of the **a** domain of wild-type (○) and PDI H55Y (●) with [GSSG] with n = 18–21 for each concentration. Panel (**D**) Relative rates of oxidation of PDI **a** domain variants with GSSG. Rates are normalized to wild-type (355 M^−1^s^−1^), with all rates determined from linear fits using five concentrations of GSSG and each concentration with n = 12–23. Panel (**E**) Linear dependence of the pseudo-first-order rate constant for reduction of the **a** domain of PDI H55Y (●) and G54P H55F (○) with [GSH]^2^ with n = 17–21 for each concentration. Panel (**F**) Relative rates of reduction of PDI **a** domain variants with GSH. Rates are normalized to wild-type (9.93 × 10^4^ M^−2^s^−1^), with all rates determined from linear fits using four to seven concentrations of GSH, each concentration with n = 12–23. All variants in panels **D** and **F** show significant difference from the wild-type (*p* < 0.001).

**Figure 3 antioxidants-11-01920-f003:**
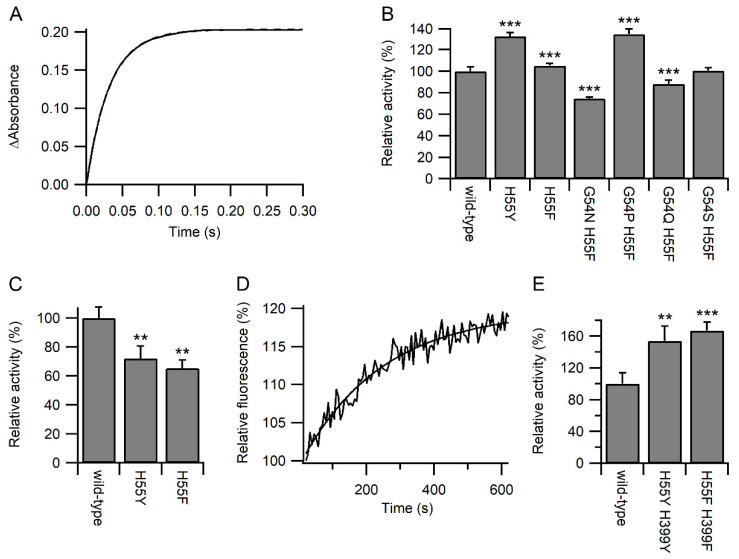
Kinetics of PDI active site reactivity. Panel (**A**) Representative plot for the reaction of wild-type PDI **a** domain with DTNB showing the increase in the absorbance at 412 nm over time. Panel (**B**) Relative rates of reaction of reduced PDI **a** domain variants with DTNB. Rates are normalized to wild-type (42.1 s^−1^); n = 18–20. Panel (**C**) Relative rates of reaction of reduced PDI **a** domain variants with DHA. Rates are normalized to wild-type (11.4 M^−1^s^−1^); n = 3–4. Panel (**D**) Representative time-dependent fluorescence profile during the deglutathionylation of the substrate peptide catalyzed by wild-type PDI. Panel (**E**) Relative rates of deglutathionylation reaction of PDI variants. Rates with the rate of non-catalyzed reaction subtracted are normalized to wild-type (0.177 min^−1^); n = 6–7. Significant differences compared to wild-type in both panels are annotated with *** for *p* < 0.001 and ** for 0.001 < *p* < 0.01.

**Figure 4 antioxidants-11-01920-f004:**
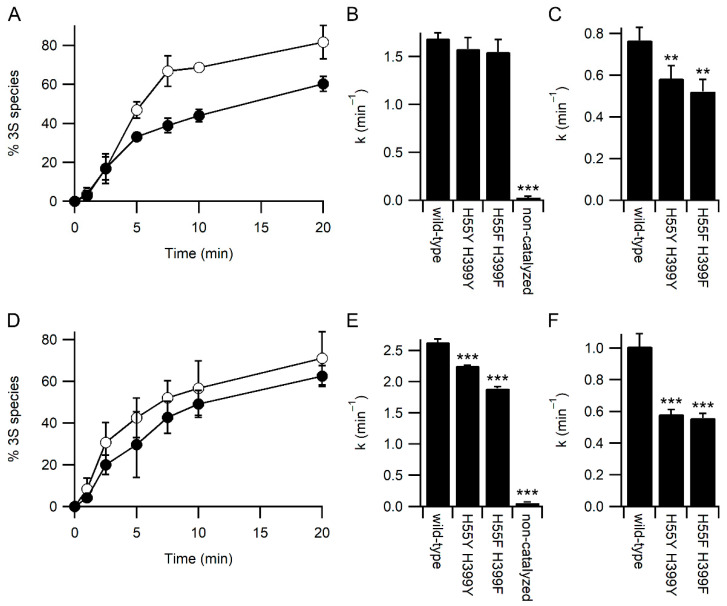
Kinetics of BPTI refolding catalyzed by PDI. Panels (**A**,**D**) Appearance of the native 3 disulfide (3S) state of BPTI catalyzed by wild-type PDI (○) and H55Y/H399Y variant (●), n = 4. Panels B and E) Rate constant for the loss of the fully reduced species (appearance of the 1S species) both catalyzed and uncatalyzed. Panels (**C**,**F**) Rate constant for the catalyzed conversion of the 1S species to the 2S species of BPTI. Panels (**A**–**C**) refolding was performed in a buffer containing 2 mM GSH and 0.5 mM GSSG. Panels (**D**–**F**) refolding was performed in a buffer containing 8 mM GSH and 1 mM GSSG. Significant differences compared to wild-type in panels (**B**,**C**,**E**,**F**) are annotated with *** for *p* < 0.001 and ** for 0.001 < *p* < 0.01; n = 4.

**Table 1 antioxidants-11-01920-t001:** Plasmids used in this study.

Plasmid Name	Construct	References
pAKL2	human PDI **a** domain (MH_6_M-D18-A137) W128F	[14]
pHIA441	human PDI **a** domain H55Y W128F	This study
pHIA440	human PDI **a** domain H55F W128F	This study
pKEHS961	human PDI **a** domain G54N H55F W128F	This study
pPK2	human PDI **a** domain G54Q H55F W128F	This study
pEDN3	human PDI **a** domain G54S H55F W128F	This study
pCS1	human PDI **a** domain G54P H55F W128F	This study
pLWRP64	human PDI (MH_6_M-D18-L508)	[15]
pHIA452	human PDI H55Y H399Y	This study
pHIA453	human PDI H55F H399F	This study
pARK32	mature BPTI (M-R36-A93)	[12]

## Data Availability

Original raw data are available from the corresponding author upon request. All materials will be made available upon request after execution of Material Transfer Agreement between institutions.

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
