# Peer review of "Introduction of a More Glutaredoxin-like Active Site to PDI Results in Competition between Protein Substrate and Glutathione Binding"

_antioxidants, 2022, doi:10.3390/antiox11101920_

Round 1

Reviewer 1 Report

This is an interesting paper from Saaranen and colleagues exploring the consequences of altering the active site of protein disulfide isomerase (PDI) to resemble a glutaredoxin.

The study shows that single mutations can shift the competition between glutathione and proteins, and the study helps towards an understanding of the evolution and diversification of thioredoxin-family proteins. The work is also noteworthy for highlighting how undergraduate practicals can be used to underpin original research in biochemistry. 

The introduction is clear but it would be helpful to expand on what is meant by “PDI has appropriate affinity” on page 2 line 53.

In the materials and methods (page 2 line 79) the authors state that “all plasmids were checked for correctness by DNA sequencing”. Is this the entire plasmid, or just the coding region? Please also define what is meant by protein “authenticity” (page 2, line 88).

I have two questions about the results:

In Figure 1, why are the UV spectra of only the H55Y variants shown and not the H55F variants?

In Figure 3, the glycine variants are explored in 3B (DTNB assay) but are not followed up for the DHA or deglutathionylation analysis in 3C-3E. Given that G54N H55F has lower activity than wt in the DNTBA assay, it would be interesting to see how it behaves in the DHA and deglutathionylation assays. Please could the authors comment.

The discussion is succinct and explains the major findings. However, I thought that the authors could have done more to map the experimental observations onto the known NMR and crystal structures of PDI. A summary paragraph to discuss the “bigger picture” at the end of the discussion would also be welcome.

On page 9, line 331 the authors state “Both the H55F and H55Y variants do show significantly increased rates of deglutathionylation catalysis (Figure 3C).” Is this not Figure 3E?

There are one or two minor spelling/grammar mistakes:

Pg 4 line 142: “it was anticipated and that these variants”

Pg 6 line 200: “To elucidate between these two options” should be “To discriminate between these two options” 

Pg 9 line 301: “via wide range of pathways” should be “via a wide range of pathways“; “via still unknown” should be “via a still unknown” 

Pg 9 line 329: “The effects of this would be predicted to be two-fold” – unclear what “this” refers to.

Reviewer 2 Report

The study by Saaranen et al. is an interesting exploration into the determinants of enzyme activity/specificity of protein disulfide isomerase, performed using an intriguing mutational approach. The paper is well organized and written, the results are clearly illustrated and convincing, and this is certainly a good article from a biochemical point of view. The biological significance of the hypothesis & results could be more clearly highlighted in the Introduction. A short "Conclusions" paragraph should also be inserted, which would significantly improve the impact of the study.
